# A Qualitative Analysis of Social-Ecological Factors Shaping Childhood Immunisation Hesitancy and Delay in the Eastern Province of Saudi Arabia

**DOI:** 10.3390/vaccines11091400

**Published:** 2023-08-22

**Authors:** Marwa Alabadi, Victoria Pitt, Zakariya Aldawood

**Affiliations:** 1School of Nursing and Midwifery, Faculty of Health and Medicine, University of Newcastle, Callaghan, NSW 2308, Australia; victoria.pitt@newcastle.edu.au; 2Primary Health Care Division of Qatif City, General Directorate of Health Affairs in the Eastern Region, Ministry of Health, Qatif 31911, Saudi Arabia; zakariyaa@moh.gov.sa

**Keywords:** public healthcare, childhood immunisation, childhood vaccination, vaccine hesitancy, delayed vaccination, Saudi Arabia

## Abstract

(1) Background: Immunisation is a crucial and effective method for preventing infectious diseases, with its success dependent on high immunisation rates to protect under-immunised individuals and promote herd immunity. This qualitative descriptive study is part of a larger explanatory sequential mixed method design that aims to explore factors influencing parents’ decision making to complete childhood immunisation in the Eastern Province of Saudi Arabia, a country experiencing disparities in immunisation coverage across its population. (2) Methods: The sample consisted of a subset of participants from the initial quantitative phase, which included a survey on the immunisation attitudes of parents living in Qatif. This initial phase included *n* = 350 participants, who were over 18, had access to one of the 27 Primary Health Care (PHC) Centres in Qatif, and had a child under 24 months. This paper presents the qualitative–descriptive phase, which used a qualitative survey to gain open-ended responses from parents (*n* = 20) and analysed using thematic analysis. (3) Results: Participants identified certain vaccines, particularly MMR, as influencing their immunisation practices. Specific factors identified as deterring parents from immunising their children included fear of autism and other developmental delays, concerns about risks and side effects, mistrust in vaccine efficacy, and discouraging information from the media. Parents’ immunisation decisions were evidently affected by policy compliance, family and friends, and social networking sites. These factors are explained through the socio-ecological model. Moreover, the COVID-19 pandemic influenced parents’ decisions on vaccine completion in terms of perceived barriers, perceived benefits, and perceived trust. (4) Conclusions: By examining the social–ecological factors shaping parents’ decisions to immunise their children in the Eastern Province of Saudi Arabia, this research contributes to the literature and informs the Saudi National Childhood Immunisation Programme about factors contributing to childhood immunisation hesitancy, helping to address a critical healthcare issue.

## 1. Introduction

Immunisation serves as a crucial and effective means of preventing infectious diseases [1]. Success is contingent upon high immunisation rates to protect the under-immunised and foster herd immunity [1]. Globally, immunisation has saved millions of lives, and as a cost-effective method for reducing disease morbidity and mortality. It is pivotal to broadening immunisation access and achieving the World Health Organisation’s (WHO) Sustainable Development Goal (SDG) 3: ensuring healthy lives and promoting well-being for all at all ages [2]. Children, whose immune systems are still developing and lack experience with pathogens, are particularly susceptible to infectious diseases [1,3].

In Saudi Arabia, immunisation campaigns since 1984 have notably decreased child morbidity and mortality from targeted diseases [4]. However, disparities in immunisation coverage persist, with some areas experiencing up to 20% non-compliance [4]. Such discrepancies raise concerns regarding the factors influencing immunisation non-compliance. A review of the existing literature, based on data from the Saudi National Childhood Immunisation Programme, reveals concerning vaccination non-compliance rates: 14.8% (57 out of 384 parents) [1], 17% (51 out of 300 parents) [5], and 20% [6]. Alshammari et al.’s research [7] found that 13% of parents neglected to complete mandatory immunisations for their children at 1 year old. Similarly, AlGoraini et al.’s study reported 14.8% of participants identified vaccine hesitancy [1]. Alsubaie et al. uncovered the statistics, with 20% of parents in their study forgoing mandatory immunisations for their children and 36% of children remaining inadequately immunised for their age [5]. The disruption caused by the COVID-19 pandemic has further intensified immunisation hesitancy and delay [1], highlighting the need to explore parents’ decision making regarding childhood vaccination in Saudi Arabia, with a particular focus on Qatif, a city in the Eastern Province.

To comprehensively examine the factors influencing parents’ decisions regarding childhood immunisation, this study adopts the social–ecological model of health-promoting behaviours [8]. This model, a variant of Bronfenbrenner’s ecological theory, delves into the multidimensionality of health-related factors across five levels: intrapersonal attributes, interpersonal processes, community aspects, institutional factors, and public policy [8,9]. Understanding the social–ecological barriers impacting parents’ childhood immunisation decisions is crucial to identifying the factors contributing to delays or refusals of immunisation. Through a qualitative descriptive approach, this study aims to shed light on parents’ decision making regarding their children’s vaccination and the complex interplay of factors influencing their choices. The study seeks to address the following research questions:RQ1. What is the parents’ decision making regarding the vaccination of their children?RQ2. How has the COVID-19 pandemic impacted parents’ attitudes and completion of childhood immunisation?

By exploring parents’ perspectives and concerns, and delving into their lived experiences and decision making, this study will contribute to both research and healthcare practice. The findings will not only fill knowledge gaps in the current literature but also enable healthcare practitioners and policymakers to develop targeted strategies to address vaccine hesitancy and improve immunisation rates. Ultimately, this research seeks to support a more robust Saudi National Childhood Immunisation Programme, enhancing the well-being of children and contributing to the overall health of communities in the Eastern Province of Saudi Arabia and beyond.

### 1.1. The Social–Ecological Model 

The social–ecological model [8] serves as a theoretical framework for understanding the intricate interplay of factors influencing human behaviour and health outcomes. This model posits that individual behaviour is shaped by multiple levels of influence and asserts that effective interventions should target these various levels [8]. Numerous dynamic factors affect parents’ decisions regarding childhood immunisation. This study proposes the use of the social–ecological model to investigate parents’ decisions surrounding childhood immunisation in Saudi Arabia’s Eastern Province. The subsequent sections will review the levels of the social–ecological model and the elements influencing parental barriers to immunisation.

### 1.2. Intra- and Interpersonal Levels

Intrapersonal factors shaping parental immunisation decisions encompass socio-demographic attributes (education, occupation, income, and race), vaccine literacy, and attitudes/beliefs about childhood immunisation [10,11]. Knowledge concerning vaccines can affect compliance to immunisation schedules. Parents or guardians lacking an understanding of immunisation benefits may either neglect to immunise their children or provide incomplete immunisation series [8]. Alongside, difficulty in obtaining accurate information concerning the recommended immunisation schedule can also result in incomplete or missed vaccinations [10,11]. Vaccine hesitancy can stem from attitudes and beliefs that vaccines harm children’s health [8,12,13], serve as government control mechanisms [8] or cause illness instead of providing protection [8,12,13]. 

Various demographic traits hinder immunisation efforts, including limited transportation to clinics, time constraints [14], and rural residence with unmet social needs, financial stress, and higher rates of chronic conditions, leading to missed opportunities for vaccination [1,15,16,17]. Low maternal education also serves as a barrier to immunisation [18], while financial constraints significantly influence uptake [19]. Additionally, an immunisation coverage disparity exists among children below the poverty level compared to those above, except for the Hepatitis B dose given at birth [20]. Parental immunisation hesitancy is observed early on, and those delaying or refusing vaccines are more likely to know someone who experienced severe vaccine reactions or parents who refused or delayed their child’s immunisation [21]. Trust in healthcare providers plays a crucial role in promoting vaccine acceptance and reducing hesitancy [22]. Common reasons for delay or refusal include concerns about vaccine side effects, autism links, and misinformation, while less common reasons include inconvenience, missed appointments, transportation issues, and financial constraints [21,23].

Additionally, the interpersonal factors influencing parental immunisation decisions include social influences, such as social norms among family and friends, and social networks. Social Contagion Theory indicates that an individual’s attitudes and behaviours can be contagious to others within their social network [23]. These social networks can positively or negatively influence individuals’ healthcare decisions [24]. Individuals in low-income communities tend to have greater social distance, resulting in less exposure to positive healthcare decisions [23,25]. In this context, mothers with a positive relationship with and trust toward their children’s doctors have more positive vaccine perceptions and are more likely to agree to immunisations [5,26], while family pressure for vaccine approval is another factor in childhood immunisations [25,27].

### 1.3. Institutional, Communal and Policy Levels

Institutional and community-level factors impact childhood immunisation. Institutional factors include relying on other parents for vaccine information, accessibility of medical facilities offering preventive services, and parents’ utilisation of these facilities for their children. Low-income populations with unmet social needs face challenges in accessing healthcare facilities, leading to lower immunisation rates [28]. Rural communities, with fewer providers and limited facilities, also struggle to access healthcare services, impacting immunisation rates [18,29,30]. Social norms perpetuate false narratives and vaccine conspiracy theories, reducing the intention to immunise [21,31]. Myths within communities, such as concerns about live viruses or autism links, contribute to negative perceptions of vaccines. Additionally, philosophical and religious reasons, as well as perceived care quality in healthcare institutions, can serve as barriers to vaccine compliance [21,32].

Immunisation policies were initially implemented to control vaccine-preventable diseases, leading to increased coverage and decreased morbidity [14,33]. However, policy-level barriers to vaccine compliance are linked to health insurance status, with low-income and uninsured individuals being less likely to be immunised [34]. 

The social–ecological model provides a comprehensive framework to understand childhood immunisation hesitancy and delay, considering factors at the intrapersonal, interpersonal, and institutional levels [34]. By examining these interconnected dimensions, targeted interventions can be developed to promote vaccine acceptance and improve public health outcomes. The study also considers the impact of the COVID-19 pandemic on parents’ attitudes and completion of childhood immunisation.

### 1.4. Vaccines in the Age of COVID-19

COVID-19-related conspiracy beliefs have influenced vaccine hesitancy and refusal. In Arab countries, the acceptance rate for the COVID-19 vaccine was 29.4%, much lower than the international average of 79.1%, with concerns about microchips and infertility impacting acceptance [35]. In the United States, 10.8% expressed vaccine hesitancy for COVID-19, citing concerns about the vaccine, a need for more information, anti-vaccine attitudes, and lack of trust [36]. Parental beliefs about vaccinating children varied, with around 27% of parents with children aged 5–11 eager to vaccinate as soon as authorised, and 31% of parents with children aged 12–17 stating they would definitely not vaccinate their child [37]. Concerns for parents of younger children included unknown long-term effects and serious side effects, with around two-thirds expressing worries about the vaccine affecting future fertility [37]. Conspiratorial thinking, influenced by individual differences, also impacts vaccine refusal, with high reactance, disgust toward blood and needles, and strong individualistic/hierarchical worldviews playing a role [26]. Trust is a significant factor in parental vaccine hesitancy, involving trust in the vaccine, healthcare providers, and policy-makers [21]. The socio-ecological model emphasises trust as critical across all levels in explaining immunisation hesitancy.

### 1.5. The Research Gap

Despite the growing recognition of the significance of childhood immunisation and the increasing attention to vaccine hesitancy, there is a noticeable gap in the current literature regarding a comprehensive exploration of the social–ecological factors that shape parents’ decisions to complete childhood immunisation, especially within the context of the Eastern Province of Saudi Arabia. While existing studies offer insights into vaccine hesitancy, they often focus on specific factors in isolation or fail to provide an in-depth understanding of the intricate interplay between intrapersonal, interpersonal, institutional, and contextual influences on parents’ decision making. Therefore, this study aims to bridge this gap by employing a qualitative descriptive approach to delve into the multifaceted determinants of childhood immunisation hesitancy and completion in this specific geographical and sociocultural context.

## 2. Methodology

This study is part of a larger explanatory sequential mixed method design project that aims to explore factors influencing parents’ decision making to complete childhood immunisation in the Eastern Province of Saudi Arabia. The initial quantitative phase included a population of 350 parents [14]. 

### 2.1. Study Design

This study used a qualitative descriptive approach to further explore parents’ decisions to complete childhood immunisation in the Eastern Province of Saudi Arabia. Qualitative research is applicable when investigating the significance of human experiences and the meaning of a phenomenon among those experiencing it [38,39,40]. Saldana [41,42] supports the use of a qualitative approach when investigating personal experiences and social interactions. This approach is more relevant when answering the “how” and “why” questions through an exploration of the lived experiences of parents who made immunisation decisions. 

The selected qualitative descriptive design is well-suited for addressing the research questions posed in the study as it allows for a clear and concise exploration of participants’ experiences, perspectives, and decision-making processes in their own words. A qualitative survey was employed, allowing participants the opportunity to express their views through open ended responses. Given the complexity and nuances involved in parents’ decision making regarding childhood immunisation, the qualitative descriptive design aligns well with the need to capture and describe the various factors influencing their choices [43]. It enables the researchers to directly collect and analyse data from the participants to gain insights into their beliefs, concerns, and motivations, providing a comprehensive overview of how parents navigate vaccination decisions for their children.

The unit of analysis for this study was parents who had to make immunisation decisions. Kumar described the unit of observation as items that are collected, measured, and observed while seeking to understand the unit of analysis [44]. An advantage of the qualitative descriptive approach is that the participants are allowed to narrate their understanding of their experiences in their own words [45]. The main foundation of this approach is the ability to generate participants’ accounts [39,46]. This will help address the research questions in this study through understanding the participant experiences of immunisation decision making.

### 2.2. Recruitment and Data Collection

The sample used in this study was a subset of participants from a previous quantitative research project (Phase 1), which explored immunisation compliance, attitudes and knowledge of parents living in Qatif. Participants includes parents aged 18 years old or over, had access to one of the PHC centres in Qatif, Saudi Arabia, and had a child that was 24 months of age or younger. After the completion of Phase 1, an email invitation was sent to 50 parents (randomly selected to avoid the subjective bias) from those who consented to participate in Phase 2. Of these, 33 responded. These 33 participants were sent an email asking them the following two questions to ensure they met the selection criteria for this study:Do you have any vaccination hesitancy regarding the childhood immunisation program?Do you live in an urban or rural area?

Participants were purposefully selected based on their responses to the first question regarding immunisation hesitancy. The criteria for selection were determined by the presence of immunisation hesitancy, which aligned with the focus of the study. This purposive sampling approach was chosen to ensure that participants possessed relevant experiences and perspectives on childhood immunisation hesitancy. The intent was to gather a rich and diverse range of insights related to the research questions. The process involved multiple rounds of sampling to refine the selection. Participants who did not meet the criteria of immunisation hesitancy were thanked for their interest and informed that they did not fulfill the eligibility criteria for participation. This iterative process allowed for the selection of participants who best represented the target group of parents with immunisation hesitancy experiences.

To prevent any potential gender-based bias, equal gender representation was intentionally maintained within the sample. This approach aimed to ensure that both male and female perspectives were adequately represented in the study, minimizing any potential subjectively gender-based discrimination. The adequacy of the sample size was ascertained through a careful balance between achieving saturation by following Guest, Namey and Chen [47] “New Information Threshold”, where new insights were no longer emerging from the data, and considering the feasibility of data collection within the research timeframe. This approach allowed us to capture a comprehensive range of perspectives while maintaining the practicality of the study. In total, 20 eligible parents who met the criteria were invited to participate in the study. Invitations, including links to the online qualitative survey, were sent via email. A reminder email was dispatched one week after the initial invitation to encourage participation and facilitate data collection.

In summary, the purposive sampling technique was employed to intentionally select participants with immunisation hesitancy experiences. The criteria for selection were determined by the research focus, and the process involved iterative rounds of sampling to refine the participant pool. The sample’s adequacy was determined through a balance of saturation and feasibility considerations. Equal gender representation was ensured to mitigate potential gender-based bias. The outcome was a well-rounded and diverse sample that contributed valuable insights to the study’s objectives. 

The qualitative survey included two closed-ended and four open-ended questions. The two closed-ended questions with limited responses, concerning participants’ immunisation status of their youngest child and identification of the immunisations they feared. The four open questions asked them to provide descriptions of their fears regarding immunisation, a description of certain events resulting in initial immunisation decisions, health-related fears associated with not vaccinating their children, and the impact of COVID-19 on the completion status of their children’s vaccination. 

This study obtained ethical approval from the University of Newcastle, Australia (ethics reference no. H-2021-0378), and the Ministry of Health of Saudi Arabia (ethics reference no. QCH-SREC07/2022). The ethical principles of autonomy and nonmaleficence have been addressed during this study. Addressing the principle of autonomy all participant signed the informed consent prior to participating in the study. Prior to consenting information sessions were conducted for prospective participants to ensure they were adequately apprised of the research purpose, procedures, and expectations to assure the voluntary, noncoercive nature of participation in the study. Participants were also notified of their right to withdraw from participation at any time. 

Addressing the principle of nonmaleficence, participants were assured that confidentiality and anonymity would be maintained. Participant identities were protected through the use of code names and no personal identifying information was collected as part of the study. 

### 2.3. Method of Data Analysis

Braun and Clarke’s [48,49] six phases for thematic analysis were utilised for qualitative coding. The first phase of the data analysis process entails familiarisation with the data. The iterative process of reading allowed for data understanding. The reading process involved reading the responses to achieve familiarity with the data and understand patterns within the data. After familiarisation with the data, initial codes were generated. Applying an inductive process, initial codes were generated as they reflected issues raised in the data. Each transcript was manually coded, extracting the thread of text and the resulting initial codes. The next phase of data analysis process involved merging data based on the similarity in patterns as per the identified initial codes to devise and determine categories among the codes [49]. The significance of assigning codes was to generate clear terms defined by the social–ecological model. The data were categorised into the dimensions of the social–ecological model during the development of the codes. The codes were then organised into patterns having similarity that were converted into sub-themes. Saldana supports reading participants’ responses and focusing on critical text when assigning codes [42]. The study utilised codes to generate code categories through a merging process. 

The third phase of the data analysis entailed widening the second-level codes into groups. This phase consisted of searching for themes through an iterative approach, going back and forth to define meaningful themes based on the theoretical framework. NVivo 12 software was used during this stage. The research assigned the second-level codes categories with short names. The researchers then reviewed the categories to determine the existing relationship among the emergent codes and create initial themes. The codes were examined for redundancy as the identification of themes continued. The codes that were not frequent or lacked similarity were discarded since they might have been coded incorrectly.

The fourth phase entailed reviewing for meaning to determine how the themes supported the research questions [48,49]. The coding process facilitated the development of the themes using the study phenomenon to provide insights into the developing themes. The extracted data were examined to determine whether they supported the themes, overlapped with the themes, and were consistent. This phase concluded when no new information existed to affect the analysis. 

The fifth phase involved assigning themes descriptively. Names of the themes depicted their relevance to other themes and the research questions. Braun and Clarke [49] affirm the importance of this step. In the current study, theme names effectively describe Saudi parents’ experiences regarding barriers to immunisation. 

The final phase entailed the interpretation of the themes to generate a narrative of the data. The summary utilised quotes from the data to support the relevance and development of the themes. The themes reflected the participants’ stories, providing a foundation for the research questions.

#### Researcher’s Characteristics and Reflexivity

As a local researcher hailing from Saudi Arabia and a doctoral student at the University, my personal and professional nursing backgrounds play a significant role in shaping the context of this study. Being intimately familiar with the cultural, societal, and contextual nuances of the Eastern Province of Saudi Arabia, where the study was conducted, has granted me a unique perspective on the intricacies that influence childhood immunisation hesitancy and delay among parents in this region. This familiarity has enabled me to establish rapport and trust with the participants more easily during the data collection process.

Furthermore, my status as a doctoral student embarking on this research as part of my Ph.D. dissertation underscores my commitment and dedication to understanding this critical issue. My academic journey has equipped me with a thorough understanding of qualitative research methods and reflexivity principles, empowering me to approach this study with methodological rigor.

In terms of reflexivity, my role as a researcher inherently brings subjectivity into the research process. I acknowledge that my background, experiences, and preconceptions could influence various stages of the study, such as data collection, analysis, and interpretation. To address this, I maintained a reflexive journal throughout the research process, documenting my thoughts, biases, and emotions as they surfaced. This reflective practice assisted me in acknowledging and critically examining my own positionality and potential biases, allowing me to make conscious efforts to minimize their impact on the study’s findings.

Furthermore, engaging in regular discussions and debriefing sessions with my research advisors and peers provided an external perspective that helped mitigate the potential effects of my subjectivity. These discussions allowed for constant self-reflection and a collaborative assessment of how my personal background may have influenced various decisions and interpretations.

In summary, the interplay between my local background, professional nursing experience, academic journey, and reflexivity practices has been instrumental in enriching the depth and contextuality of this study. I have made earnest efforts to balance my subjectivity with methodological rigor, and I believe that this reflexivity enhances the transparency and credibility of the research findings.

## 3. Results

This section describes the analysis of the collected data based on the research questions. The themes generated from the thematic analysis of the data are presented under the two research questions that guided this study.

RQ1.What is the parents’ decision making regarding the vaccination of their children?

The first research question focused on parents’ decision making regarding the vaccination of their children. Table 1 includes the demographics of the participants, such as age, gender, and socioeconomic background while Table 2 presents the vaccines cited by parents as having determined their decision to refrain from immunising their children. Participants highlighted the greatest fear around the MMR vaccine, which 10 of them (50%) cited was the primary vaccine they feared. However, a variety of other immunisations were also named.

The themes that emerged indicated several factors that influenced this decision making and was directly related to the five levels of the socio-ecological model (i.e., intrapersonal, interpersonal, institutional, community, and policy). These factors are discussed below.

### 3.1. Theme 1. Interpersonal and Intrapersonal Factors

Reflecting the interpersonal and intrapersonal factors, the emergent themes were associated with concerns regarding fear of autism, risks and side effects, and discouraging information from the media. For instance, one parent mentioned how the fear began to develop:

“*My fears started to develop when I first got pregnant and started to read about them on the Internet. A lot of the information I read talks about the side effects that these vaccines cause. Most of them state that a lot of children have real reactions to allergens in vaccines*.” 

The parents commented that they feared that their children could suffer disability or permanent injury. The most reported fears were compromised immune systems, allergies, autism spectrum disorder, and other developmental disorders. The fears were associated with distrust in the vaccine and beliefs that immunisation efficacy is overstated. 

According to one parent: 

“*Based on my research during my last pregnancy about childhood immunisations, I have some concerns regarding the side effects of some of the vaccines, including the possibility of language delays, direct and indirect causes of autism, and the risk of immune compromise associated with unnatural ways of gaining immunity*.” 

Another participant said the following:

“*I’m a health professional, and I honestly have some personal fears regarding MMR doses. Several researchers from around the world have concerns about the side effects of MMR and how it could cause autism, different reactions to allergens in vaccines, and long-term, as of yet unknown, side effects*.” 

Table 3 summarises specific fears cited by parents, were their children to have been immunised. Eight parents (40%) cited that they feared autism, seven participants (35%) had concerns of risks and side effects of vaccines, seven parents (35%) received discouraging information from the media, and three of them (15%) indicated that they had fears of exposing their children to toxic ingredients.

### 3.2. Theme 2: Institutional, Community, and Policy Factors

The results highlight institutional, community and policy factors as information sources that affected the immunisation practices among parents. As shown in Table 4, the most cited sources of information regarding immunisation included policy compliance (11), family and friends (8) and social networking sites (6). These factors reflect the policy, institutional, and community levels within the socio-ecological model. The participants described how compliance with local regulations forced them to immunise their children during the first months of life. Participants also recalled that family members and friends helped them decide to immunise their children. Some parents mentioned that they heard “a lot on social media about how MMR results in some side effects for chronic conditions such as autism”. Another participant cited “compliance, family, Internet sites sponsored by the Ministry of Health, printed brochures and TV materials on childhood immunisations, friends, instinct and personal feelings” as crucial factors influencing their decision making. Other cited factors included instinct, feeling or suspicion, institutional support, support groups, and television news programmes. 

RQ2.How has the COVID-19 pandemic impacted parents’ attitudes and completion of childhood immunisation?

The final research question focused on how the COVID-19 pandemic affects parents’ decisions regarding childhood immunisation completion. As shown in Table 5, the emergent themes were categorised into perceived barriers (8), perceived benefits (3), and perceived trust (9). 

Parents perceived COVID-19 as a barrier to completing immunisations. One parent described that: “*COVID helped me stand up for my decision due to the number of cases of those who took the three shots and suffered from the side effects*”. Another parent reflected that vaccine producers proved to the world that they care about nothing except money, as so many people died because of COVID-19 vaccines. These narratives demonstrate parents’ beliefs that vaccines should not be trusted. However, not all participants agreed, as one participant argued: “*It didn’t affect me at all as I took the COVID vaccines, and I didn’t experience any side effects*”.

## 4. Discussion

This qualitative descriptive research aimed to describe the parents’ decisions to complete childhood immunisation in the Eastern Province of Saudi Arabia. This study has identified the social–ecological factors contributing to parents’ decisions to complete childhood immunisation. A noticeable gap was identified in the existing literature pertaining to the identification and role play of social–ecological factors within the context of the Eastern Province of Saudi Arabia. Although the existing body of knowledge addresses the factors behind immunisation hesitancy and delay, yet the interplay among intra- and interpersonal, institutional and contextual factors on parents’ decision making are rarely attributed and addressed. The findings align with McLeroy et al.’s social–ecological model [8], which defines individuals’ health-related factors in the context of their intrapersonal attributes, interpersonal processes, community factors, institutional factors, and public policy. The interpersonal and intrapersonal attributes identified in this study that defined parental immunisation decisions revolved around parental fear, vaccine literacy, and parental disbeliefs in childhood immunisation. These factors specifically included autism beliefs, concerns about risks and side effects, discouraging information from the media, and fear about exposure to toxins in vaccines. Parents or guardians who do not understand the benefits of immunisation do not start or do not complete their immunisation series [13,27]. Trust in the healthcare provider was reported as the main reason not to delay a vaccine or refuse a vaccine when a parent had been considering doing so, which demonstrates the crucial role healthcare providers play in promoting vaccine acceptance and reducing vaccine hesitancy [13,27]. 

Institutional and community factors identified in this study as influencing parental immunisation decisions include social influences such as social norms among family and friends and social networks [27]. These social networks can positively or negatively influence individuals’ healthcare decisions [13]. The effect of policy was also indicated by the participants. Importantly, trust in vaccine recommendations communicated by the authorities, social networks, and friends and family can have a significant impact on parents’ trust level and, hence, their decision on whether to immunise their children [50].

The participants’ responses highlighted fear of certain vaccines, particularly MMR. The parents described specific fears and concerns related to immunisation, such as autism, risks and side effects, identifying discouraging information from the media, and exposure to toxins in vaccines. Parents in the current study feared that their child’s immune system was not strong enough to handle the vaccines. These results are similar to those of Costa-Pinto et al. [51] and Enkel et al. [52], who found parental fears around vaccine safety. However, one parent did not fear their child being exposed to a disease, noting that it was preferable that their child acquire the disease than receive immunisation, and that diseases boosted children’s immunity. Some parents considered that some vaccines were no longer relevant and believed that their children were not at risk of contracting these diseases. These results are in line with Hough-Telford et al.’s [50] findings on parents’ refusals to immunise their children. The findings demonstrate that belief in medical conspiracy theories affects parents’ decisions to immunise their children as parents refused immunisation due to this belief. Individual differences that have been found to impact anti-immunisation stances, conspiratorial thinking, and vaccine refusal are high reactance, high levels of disgust toward blood and needles, and strong individualistic/hierarchical worldviews [50]. 

The parents in this study cited various information sources that affected their immunisation practices. The participants described how compliance with local regulations forced them to immunise their children during the first months of life. This compliance with Saudi Arabian policy relates specifically to the policy level within the social–ecological model [5]. Immunisation regulations are implemented to increase community immunity and childhood immunisation coverage, thereby safeguarding the general public and individual health. These findings demonstrate that parents’ trust in government as a facilitator for immunisation. Ullah et al. [53] found that individuals who delay or refuse vaccines are more likely to have searched for vaccine information online. The parents indicated that family and friends were also a motivation in adhering to the recommended schedule and immunising their children. 

Parents cited media as a source of information and motivation to understand the risks involved with immunisation decisions. Media such as social media, television programmes, and websites informed parents about the likely dangers of immunisation and evoked emotions of fear [54]. These findings demonstrate that media has an influence on parents’ decision making; without media reports on vaccines, some parents may not question immunisation. In the current study, media resulted in negative feelings, although media has been found to have a positive effect. Gautam [54] found that 96% of parents cited that they undertook polio immunisation after observing immunisation campaigns on television. Ullah et al. [53] found that children from households who had access to media had received a higher number of vaccines than children from households without electronic media. 

Trust appeared to be a crucial factor in parental vaccine hesitancy, with vaccine acceptance requiring parents to have trust in the product, provider, and policy-makers [13,27]. The parents considered whether they trusted the information itself and trusted the people who produced and propagated the immunisation information. In essence, trust in vaccine information itself is nested within and dependent upon trust in the source of information [16]. The perceived trustworthiness of information itself and the source of information is subjective as it depends on a person’s personal experiences and biases [13,27]. 

The parents demonstrated a strong sense of mistrust in institutions and healthcare as a result of the COVID-19 pandemic. Mistrust towards immunisation was expressed as suspicion, and some parents felt that COVID-19 affirmed their decision due to the number of cases where people took three shots and suffered from side effects. They also expressed that vaccine producers did not care about anything except money. The previous literature on the COVID-19 vaccine, as well as vaccines in general, has shown how trust of various kinds impacts vaccine hesitancy [21,25]. The results of this study further emphasise the important role of trust in childhood immunisation Figure 1 presents the thematic model of the social–ecological factors shaping childhood immunisation hesitancy and delay in this study.

## 5. Research and Practical Implications

The findings in this study demonstrate that there are social and ecological factors that influence childhood immunisation and immunisation hesitancy among parents in Saudi Arabia. 

It is evident that different measures are needed to counter the social–ecological aspects of vaccine hesitancy. In situations where parents demonstrate concerns about risks and side effects, healthcare practitioners can offer perspectives and discuss with parents about risks and benefits through social events. However, since this situation arises due to a lack of trust in clinicians and researchers, there is another dimension to the interaction. Healthcare practitioners can manage this by building trust within communities and explaining the rigour of research governance to avoid conflicts of interest. While studies cite immunisation as an effective public health approach for preventing mortality and morbidity of vaccine-preventable diseases, healthcare practitioners find it a challenge to discuss the benefits and risks of immunisation with parents. This is a concern in Saudi Arabia since the rates of completing immunisation vary across the regions, with some areas reporting an immunisation non-compliance rate of up to 20%. Understanding the social–ecological factors contributing to immunisation hesitancy is critical to this vulnerable population. 

Saudi parents who are COVID-19 vaccine hesitant were not monolithic. COVID-19 vaccine hesitancy comes from a combination of factors, which appear to make up unique and distinct archetypes of vaccine-hesitant parents. The current study’s preliminary findings of heterogeneity amongst vaccine-hesitant parents are consistent with research identifying different vaccine-hesitant archetypes for the COVID-19 vaccine [50]. With vaccine-hesitant archetypes established, future research can investigate the specific mindset, attitudes, beliefs, and influences of these archetypes and then tailor engagement and messages to each archetype based on these findings. While the current study demonstrated that trust in information sources and aspects of the social–ecological model contribute towards vaccine hesitancy, future research should also investigate tailoring messages and engagement around trusted sources and different constructs of the social–ecological model to see if that will improve immunisation acceptance and parents’ decisions to immunise their children. Future research should explore whether targeting specific constructs of the social–ecological model and specific trust constructs could be useful in changing attitudes in parents who fit certain vaccine-hesitant archetypes.

## 6. Limitations

This study has several limitations. The investigation was carried out in a single city, only focusing on parents in Qatif with 20 participants, which limits the generalizability of the study results to alternative environments in Saudi Arabia. The participants recruited in this study had children that were of 24 months or younger, which might not have accounted for the potential differing immunisation views and practices of parents with children older than 24 months. Another limitation of the study might lie in the selected data collection tool, i.e., the survey method instead of semi-structured in-depth interviewing. All the participants from Phase 1 were in touch through an online mechanism, which is why it was easier for the researcher to utilise the online semi-structured questionnaire. This method helped this study find and access the relevant and potential participants for the study, which might not have been possible through in-person and face-to-face interviews. Pertaining to the current limitation, another study might be conducted by opting for the in-person interview method and the results might be compared for further clarification. Future research should extend these findings to other regions of Saudi Arabia. In addition, future research should consider developing interventions to approach the reported social–ecological factors contributing to vaccine hesitancy.

## 7. Conclusions 

This qualitative descriptive research aimed to explore parents’ decisions to complete childhood immunisation in the Eastern Province of Saudi Arabia. The study reveals complex and multifaceted parental behaviours regarding immunisation decisions, driven by the desire to do what is best for their children. Three themes emerged from the data analysis, shedding light on the factors shaping parents’ decisions. These themes were interpersonal and intrapersonal factors, institutional, community, and policy factors, institutional, community, and policy factors. 

Since these behaviours are founded on the need to do what is considered right for their children, this study’s findings can be applied by healthcare practitioners when approaching education and support of a parent. Specific fears that deterred parents from immunising their children included concerns about autism, risks and side effects, negative information from the media, and apprehension regarding exposure to toxins in vaccines. Parents in this study cited various information sources affecting their immunisation practices, with media playing a significant role. The COVID-19 pandemic led to a strong sense of mistrust in institutions and healthcare among parents, which in turn impacted their immunisation decisions. Consequently, this study contributes to the literature by exploring the social–ecological factors shaping parents’ decisions to immunise their children in the Eastern Province of Saudi Arabia. Public health professionals can utilise this study to identify parents who opt to refuse immunisation. To implement targeted interventions within this population, it is crucial to consider the lived experiences and perceptions of non-compliant parents in order to pinpoint the root causes of immunisation refusal.

The study’s findings underscore the importance of understanding parents’ lived experiences and perceptions to identify the root causes of immunisation refusal. Health practitioners can use this knowledge to tailor targeted interventions and support non-compliant parents. By considering these themes, public health professionals can effectively address concerns and misconceptions, promote vaccine acceptance, and improve childhood immunisation rates in the region.

## Figures and Tables

**Figure 1 vaccines-11-01400-f001:**
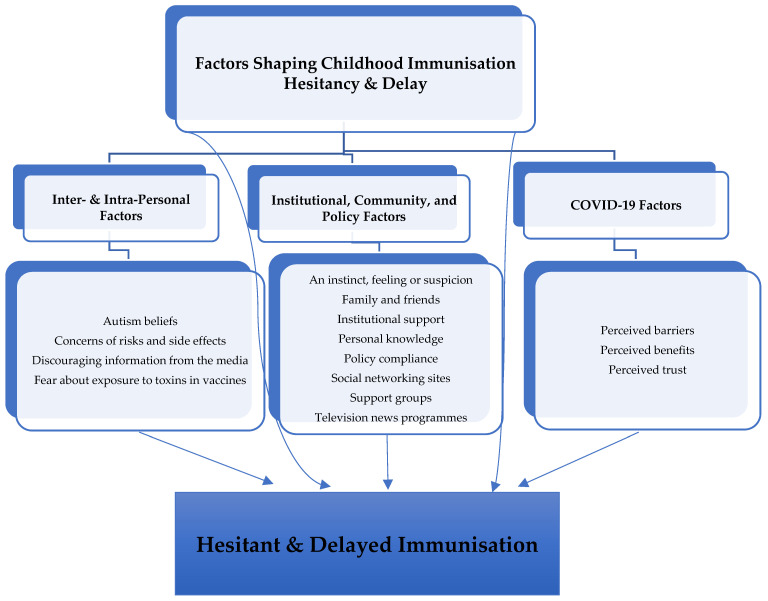
Thematic Model.

**Table 1 vaccines-11-01400-t001:** Demographics of the participants.

Gender	Frequency	Age Group	Frequency	Socio-Econ. Status	Frequency
Males	10 (50%)	26–35	8 (40%)	Upper	3 (15%)
Females	10 (50%)	36–45	7 (35%)	Middle	17 (85%)
		45 and above	5 (25%)	Lower	00 (00%)
Total	20 (100%)		20 (100%)		20 (100%)

**Table 2 vaccines-11-01400-t002:** Vaccines cited as the main factor in changing immunisation practices.

Vaccines	Frequency
All vaccines	7
Bacille Calmette–Guérin	3
Diphtheria, pertussis and tetanus toxoid	4
Hepatitis B vaccine	6
I do not fear any vaccines	2
Mumps, measles and rubella vaccine (MMR)	10
Oral polio vaccine	4

**Table 3 vaccines-11-01400-t003:** Interpersonal and intrapersonal factors.

Interpersonal and Intrapersonal Factors	Times Cited	Samples of Original Transcripts
Autism beliefs	8	I read a lot about the MMR vaccine, which may trigger autism before children grow up. The MMR is the primary immunisation which makes me question the safety of the given vaccines to children. I read a lot about the MMR vaccine which may trigger autism before children grow up.
Concerns of risks and side effects	7	I fear that my child will not gain natural immunity.
Discouraging information from the media	7	My fears started to develop when I first got pregnant and started to read about them on the Internet. A lot of the information I read talks about the side effects that these vaccines cause. Most of them state that a lot of children have real reactions to allergens in vaccines.
Fear about exposure to toxins in vaccines	3	I fear that the immune system of my children will be compromised by these toxic ingredients injected into their bodies at a young age

**Table 4 vaccines-11-01400-t004:** Institutional, community, and policy factors.

Institutional, Community, and Policy Factors	Times Cited	Samples of Original Transcripts
An instinct, feeling or suspicion	3	I have concerns about vaccine safety, including immediate side effects and increased susceptibility to chronic conditions such as autism.
Family and friends	8	Family members also helped me decide to immunise my child in the very first months
Institutional support	1	The hospital gives the first shots when the child is born and before leaving the hospital.
Personal knowledge	2	Researching and reading lots of online articles in this regard
Policy compliance	11	Compliance with the local regulations forced me to immunise my child in the very first months. Compliance, family, Internet sites sponsored by the Ministry of Health, printed brochures and TV materials on childhood immunisations, friends, instinct and personal feelings to protect my children the best way I could
Social networking sites	6	I heard a lot on social media about how MMR results in some side effects for chronic conditions such as autism. Internet sites sponsored by organisations that oppose current immunisations were the most frequently cited information source for me.
Support groups	3	Support groups made me feel comfortable to comply with the immunisation schedule
Television news programmes	2	TV materials on childhood immunisations influenced me to immunise my child

**Table 5 vaccines-11-01400-t005:** COVID-19 factors.

COVID-19 Factors	Times Cited	Samples of Original Transcripts
Perceived barriers	8	COVID helped me stand up for my decision due to the number of cases of those who took the three shots and suffered from the side effects
Perceived benefits	3	I took the COVID vaccines, and I didn’t experience any side effects.
Perceived trust	9	Vaccines producers proved to the world that they care about nothing except money, as so many people died because of COVID vaccines

## Data Availability

The online questionnaires are hosted in LimeSurvey (https://www.limesurvey.org/en-au/privacy-policy (accessed on 2 July 2023)) supported by the University of Newcastle, Australia. Additionally, all data collected will be stored on the University of Newcastle’s Cloud secure server for a minimum of 5 years and will only be accessible to members of the research team. Data will be securely destroyed in line with UON policy provisions.

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
