# Peer review of "A Qualitative Analysis of Social-Ecological Factors Shaping Childhood Immunisation Hesitancy and Delay in the Eastern Province of Saudi Arabia"

_vaccines, 2023, doi:10.3390/vaccines11091400_

Round 1

Reviewer 1 Report

The study “A Qualitative Analysis of Social-Ecological Factors Shaping Childhood Immunisation Hesitancy and Delay in the Eastern Province of Saudi Arabia” holds significant importance as it provides valuable insights into the social and ecological factors influencing childhood immunization and immunization hesitancy among parents in the Eastern Province of Saudi Arabia. By exploring parents' perspectives, concerns, and information sources, the study contributes to a deeper understanding of the complex decision-making processes surrounding childhood immunization. The findings can inform healthcare practitioners and policymakers in developing targeted interventions and educational initiatives to address vaccine hesitancy and improve immunization rates. Additionally, the study highlights the impact of the COVID-19 pandemic on parental attitudes and trust in institutions, emphasizing the need for tailored approaches during public health crises. The authors should address the following considerations before the acceptance for the publication.

The abstract briefly mentions the sample consisting of a subset of participants from a Phase 1 survey conducted in Qatif. However, it lacks essential information about the demographics of the participants, such as age, gender, and socioeconomic background, which could impact the generalizability of the findings. The abstract mentions thematic analysis as the method used for data analysis, but it lacks information about the data collection process, such as the type of data collected (interviews, surveys), data collection instruments, and the overall study design. Providing these details would give readers a better understanding of the rigor and validity of the study.

The introduction should be concise and focused, providing a brief overview of the background, research objectives, and the significance of the study. Long paragraphs or excessive details can overwhelm readers and detract from the main points. Prioritize the inclusion of key information that is directly relevant to the study. Avoid unnecessary elaboration or extensive literature review in the introduction. Save detailed discussions and extensive supporting evidence for the subsequent sections of the manuscript.

The methodology section lacks a clear and concise explanation of the qualitative descriptive approach used in the study. It would be beneficial to provide a brief overview of the rationale for choosing this approach and how it aligns with the research objectives. The description of the data collection process is limited. It would be helpful to provide more information about the specific methods used for data collection, such as interviews or focus groups, and how the participants were recruited. Additionally, details on the data collection instruments, interview protocols, or discussion guides should be included to ensure transparency and replicability. Furthermore, it does not mention data saturation, which is a crucial aspect of qualitative research. Data saturation refers to the point at which no new information or themes emerge from the data. It is important to discuss how the researchers determined that data saturation was achieved and ensured that enough data were collected to address the research questions adequately.

The methodology section briefly mentions ethical principles of anonymity, confidentiality, and privacy, but does not provide sufficient detail on the ethical considerations taken into account. It is essential to include information on how informed consent was obtained, how participant identities were protected, and how ethical guidelines were followed throughout the study.

The description of the data analysis process is relatively brief and lacks sufficient detail. It would be beneficial to provide a more comprehensive explanation of the steps involved in thematic analysis, including how codes were generated, how themes were developed, and how data were reviewed for meaning and interpretation. Additionally, providing information on intercoder reliability or the use of independent coders would enhance the transparency and rigor of the analysis process.

In the results section, the observation have been describe as raw results. It would be more appropriate to present the data in the form of percentages and 95% confidence intervals. Moreover, in the table 1. adjusted odd ratios should be be calculated regarding the polychotonomus variables.

The conclusion does not effectively summarize the main findings of the study. It briefly mentions that social and ecological factors influence childhood immunization and hesitancy among parents in Saudi Arabia, but it does not provide a clear overview or synthesis of the key findings or themes identified in the study.

Minor editing of English language is required

Reviewer 2 Report

Understanding reasons for vaccine, hesitancy among parents, is important to optimise vaccine uptake. This is an important and timely enquiry.

The aim of the study is to understand the reasons for vaccine, hesitancy in a region of Saudi Arabia. The method used was a qualitative approach, with which I’m not familiar, and hence I checked the source documents describing this approach that were cited by the authors. My understanding of the approach is as follows: Participants were invited to answer two questions in their own words:

“Do you have any vaccination hesitancy regarding the childhood immunisation program? 

Do you live in an urban or rural area? "

If the study subjects said they had hesitancy to accept vaccination, they were selected for further study. The selected individuals were sent a further set of questions. However, the authors have not provided the precise wording or format of the questions for me to judge the context and nature. It would be essential to provide this as an online appendix to the paper.

The answers were then studied, and patterns of responses identified and summarised.

Without access to the precise questions that were asked, and the parameters of the answers that were sought I could not understand the precise methodology used.

The results that were presented, and their analysis provided useful information and insights into the decision-making process that parents undertook to decide on whether to vaccinate the children or not. They provide guidance on how to formulate a more comprehensive and systematic study that could be applied to investigate factors contributing to vaccine hesitancy in Saudi Arabia.

However, I have significant concerns about the generalisability of these results:

1. The small size of the sample and the method of choosing the sample, increases the likelihood of ascertainment bias. By excluding all those who said they did not have vaccine hesitancy, the study has eliminated a very useful control group for comparison.

2. When is such a small group is selected for answering the aims of the study, I have real concerns that the broad demographics of the population are not covered. How do the demographics of the subjects selected for the study, compare with the demographics of the total  the population in this geographic  area?

3. I do not know how that themes for analysis, based on the answers from the study subjects, were selected and also, who did the selection? Did different individuals carry out the selection, and was there reproducibility if the theme selection carried out by different individuals was compared?

Reviewer 3 Report

This is an interesting and relevant study.  These are some suggestions:

1.  Novelty: You cite extensive literature in both the background and discussion and explain the importance of research in the field.  However, it is not clear  what the gap in the literature is that your study is filling and what your study has added to previous literature.  I suggest making this more explicit in the background and discussion sections.

2.  Ssmpling: Can you explain how you chose which 50 of the original study participants to ask to complete this additional survey.

3.  Study design: I was interested in why you chose a survey, rather than e.g. semi-structured interviews.  The latter may have produced more in depth responses.  You may want to explain this in the methods.  This is also a limitation that should be acknowledged in your discussion.

4.  DAaa analysis: You have explained the thematic analysis process in detail.  Can you explain how the different researchers worked together to ensure reliability and validity?  Many of the themes in the results section appear to overlap and it would be good to understand more about how decisions were made about which. data fitted which theme. 

5.  Structure:  Some new results e.g. regarding lack of necessity are presented in the discussion section that I could not find reference to in the results section.  Please check that you have presented any results discussed in the discussion section in the results section first.

6.  Language.  The term adherence is now preferred to compliance as it is a less judgmental term.  Similarly terms such as 'neglect' and 'alarming' have  judgmental connotation and should be avoided.

Good

Author Response

Please take a look at the attachment with all the responses. The latest changes are made in blue in the manuscript.  

Reviewer 4 Report

Summary - I have read the manuscript with interest. This qualitative study describes the factors affecting parents' decisions regarding immunization for their children. The study design is appropriate, the theoretical framework is clearly presented and the methodology is adequately described. Below are suggestions for improvement. 

1. Title - The study title can be improved. The subject of the study - i.e. parents with children upto 24 months of age, is not clear in the study title. 

2. Introduction - The background and context of the study are well described however the description of the knowledge gap that this study hopes to fill can be improved. What does this study hope to answer that has not previously been answered by other studies on parenteral hesitancy for childhood immunization?

3. Methodology - Description of researcher characteristics and reflexitivity is missing and must be added. 

4. Methodology - Survey used should be included as a supplement to the manuscript. The process of developing the survey questions should be elaborated. How did existing literature inform the survey design? How did the theoretical framework of this study inform the survey design? How was the survey design piloted, modified, validated, or otherwise changed in the design process?  

5. Sampling needs some clarification - Line 213 states "After the completion of Phase I, an email invitation was sent to 50 parents..... " How were the 50 selected? 

6. Line 219 suggests purposive sampling. This needs more elaboration. What were the criteria selected for and how were these criteria chosen? How many rounds of sampling were conducted? How was adequacy of sample ascertained? 

7. Results  - I am confused about the categorization of the themes here. Authors have categories "Discouraging information from the media" as Interpersonal and Intrapersonal factor but "Personal knowledge" and "An instinct, feeling or suspicion" has been categorized as Institutional, community and policy factor. Please clarify this. 

8. Discussion - Line 384 - "Trust in the healthcare provider was reported as the main reason ...." It is unclear where this was reported. If it was in the present study, then this is not presented in the results. If it has been reported in prior studies then past perfect tense may be more appropriate here. 

9. The thematic model presented is not well supported and needs much improvement. I would suggest that in this exploratory phase of the study, a thematic model is not needed and can be eliminated from the current presentation. 

Minor editing is required to improve clarity. 

Author Response

(The authors gave the same response as above.)

Round 2

Reviewer 1 Report

The authors have sufficiently improve the manuscript. However, it is further suggested that the frequency distribution regarding the gender, age and socioeconomic status in the table 1 show be presented in the form of percentages (95% confidence interval). Furthermore, it would be much appropriate to calculate the adjusted odd ratios regarding these variables. I would like to suggest these minor changes before accepting the manuscript for publication. 

Author Response

(The authors gave the same response as above.)

Reviewer 4 Report

I thank the authors for their careful consideration of the comments. The manuscript has been revised and I have no major concerns at this time.